# Multidecadal and climatological surface current simulations for the southwestern Indian Ocean at 1/50° resolution

Noam Vogt-Vincent[1] and Helen Johnson[1]

[1]Department of Earth Sciences, South Parks Road, University of Oxford, Oxford, UK

**Correspondence:** Helen Johnson (helen.johnson@earth.ox.ac.uk)

**Abstract.** The **W**estern **Ind**ian Ocean **S**imulation (WINDS) is a regional configuration of the Coastal and Regional Ocean Community Model (CROCO) for the southwestern Indian Ocean. WINDS has a horizontal resolution of $1/50°$ ($\sim$2 km) and spans a latitudinal range of 23.5° S – 0° N, and a longitudinal range from the East African coast to 77.5° E. We ran two experiments using the WINDS confuguration: WINDS-M, a full 28-year multidecadal run (1993–2020); and WINDS-C, a 10-year climatological control run with monthly climatological forcing. WINDS was primarily run for buoyant Lagrangian particle tracking applications, and horizontal surface velocities are output at a temporal resolution of 30 minutes. Other surface fields are output daily, and the full 3D temperature, salinity, and velocity fields are output every 5 days. We demonstrate that WINDS successfully manages to reproduce surface temperature, salinity, currents and tides in the southwestern Indian Ocean, and is therefore appropriate for use in regional marine dispersal studies for buoyant particles, or other applications using high-resolution surface ocean properties.

## 1   Introduction

The western Indian Ocean is a relatively data-sparse region. Surface current data are required to simulate the dispersion of buoyant particles such as marine debris or coral larvae (van Sebille et al., 2018), and whilst global products exist that cover the southwestern Indian Ocean, derived from satellite altimetry (e.g. Rio et al., 2014) and global ocean reanalyses (e.g. Lellouche et al., 2021), these products are at a coarse resolution relative to the scales of larval dispersal and do not resolve sub-mesoscale dynamics which are thought to be important for larval transport (e.g. Monismith et al., 2018; Dauhajre et al., 2019; Grimaldi et al., 2022). Some higher-resolution models have been run in the southwestern Indian Ocean, but these simulations only spanned a limited subset of coral reefs within the region, and are not available on publicly-accessible repositories (Mayorga-Adame et al., 2016, 2017; Miramontes et al., 2019). Our objectives were to (1) provide improved estimates of regional surface currents across the tropical southwestern Indian Ocean, including at sub-mesoscale, and (2) estimate the connectivity (and temporal variability of connectivity) of coral reefs across the region, including the Chagos Archipelago. Bridging the gap between the fine-scale dynamics that dominate in coastal seas, and large-scale ocean currents and mesoscale variability in the high seas, is a major challenge in modelling larval dispersal (Edmunds et al., 2018). Future developments in unstructured ocean models, and improvements in the availability of computational resources, will be invaluable in addressing these challenges. However, for this study, we used a $1/50°$ ($\sim$2 km) configuration of a regional (structured) ocean model to simulate circulation

in the southwestern Indian Ocean, which we call the **W**estern **Ind**ian Ocean **S**imulation (WINDS). Here, we provide a full description of WINDS and the two experiments we ran using the configuration, and validate WINDS as relevant for buoyant Lagrangian particle tracking applications.

## 2  Methods

### 2.1  Numerics

We ran WINDS using version 1.1 of the Coastal and Regional Ocean Community Model (Auclair et al., 2019; Jullien et al., 2022), coupled with the XIOS2.5 I/O server for writing model output (https://forge.ipsl.jussieu.fr/ioserver). WINDS uses a nonlinear equation of state (Jackett and Mcdougall, 1995; Shchepetkin and McWilliams, 2003) with a $3^{\mathrm{rd}}$-order upstream biased scheme for lateral momentum advection, a split-and-rotated $3^{\mathrm{rd}}$-order upstream biased scheme for lateral tracer advection, a $4^{\mathrm{th}}$-order compact scheme for vertical momentum advection, and a $4^{\mathrm{th}}$-order centered scheme with harmonic averaging for vertical tracer advection. Lateral momentum mixing is achieved through a Laplacian Smagorinsky parameterisation (Smagorinsky, 1963), and a Generic Length Scale $k-\epsilon$ scheme is used for vertical mixing (Jones and Launder, 1972). A bulk formulation is used for surface turbulent fluxes (COARE3p0) with current feedback enabled (i.e. momentum input from wind stress is relative to surface currents). The configuration uses radiative boundary conditions for forcing at the lateral boundaries (including non-tidal and tidal SSH, barotropic tidal currents and baroclinic non-tidal currents, and temperature and salinity). A 10-point cosine-shaped sponge layer is also used at the lateral boundaries for tracers and momentum. Bottom friction is implemented using quadratic friction with a log-layer drag coefficient, with $z_{0,b} = 0.02$ m, and $0.002 \leq C_d \leq 0.1$ (limits chosen for numerical stability). We used a baroclinic timestep of 90 s, with 60 barotropic steps per baroclinic step (Shchepetkin and McWilliams, 2005).

### 2.2  Model grid

We built the model grid using CROCO_TOOLS with longitudinal limits $34.62°$ S – $77.5°$ E, latitudinal limits of $23.5°$ S – $0°$ N, and a specified horizontal resolution of $1/50°$ (Figure 1). The western boundary of the domain is entirely land (East Africa). We chose this domain as it spans almost all coral reefs in the southwestern Indian Ocean, as well as the Chagos Archipelago, allowing connectivity between the Chagos Archipelago and the rest of the southwestern Indian Ocean to be investigated. A small number of coral reefs southward of $23.5°$ S (in southernmost Madagascar and South Africa) are therefore excluded, but this was necessary to keep computational and storage requirements tractable. To maintain roughly even dimensions of grid cells across the domain, CROCO_TOOLS adjusts the meridional resolution of cells away from the equator, so the true meridional resolution of grid cells at the southern boundary of the WINDS domain is slightly finer, at around $1/55°$. The horizontal resolution of WINDS is therefore approximately 2 km, but actually ranging from 2.04 km at the southern boundary, to 2.22 km at the equator.

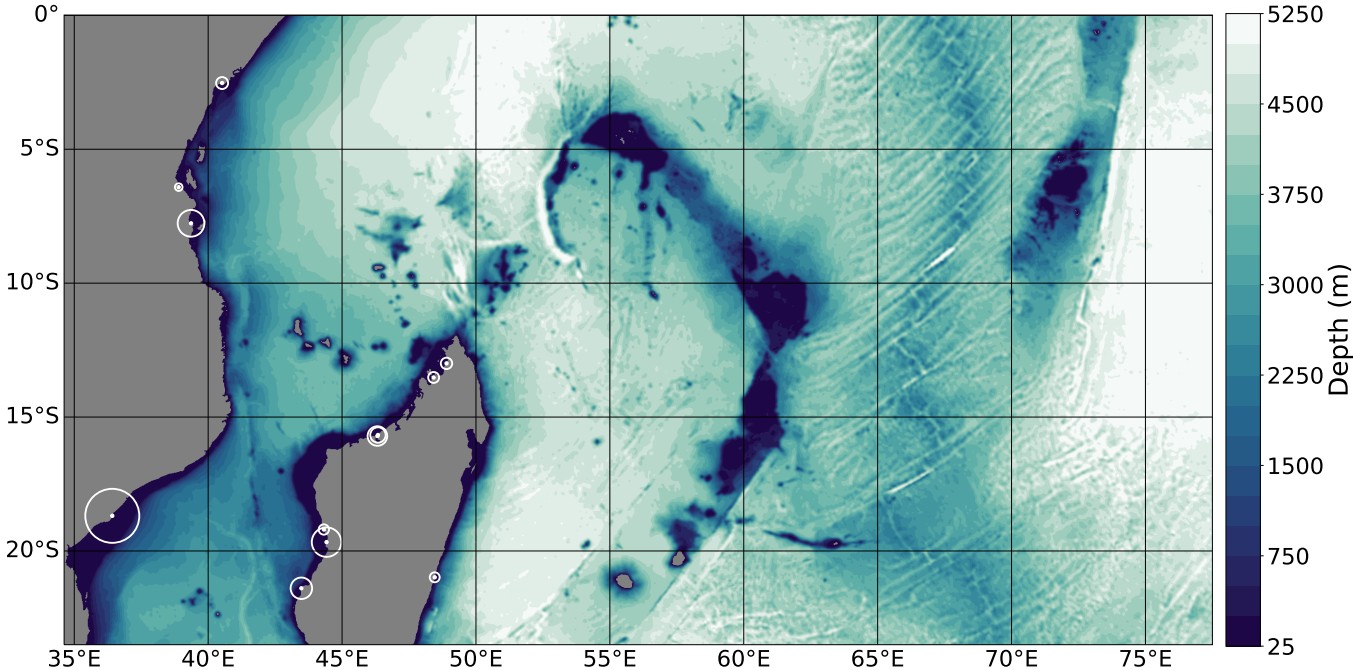

**Figure 1.** The entire WINDS domain, with contours representing the bathymetry used in WINDS. Circles represent the 12 rivers included in WINDS, scaled by the total annual discharge (Dai and Trenberth, 2002).

CROCO uses a terrain-following (s-coordinate) grid in the vertical. We used 50 vertical layers in WINDS, using a vertical stretching scheme that improves the resolution at the surface and bottom boundary layers, defined by the parameters $\theta_s = 8$, $\theta_b = 2$, and $h_c = 100$m (see the CROCO documentation for the technical explanation of these parameters). Since s-coordinates are terrain (and sea-surface) following, translating s-coordinates to depth depends on the local ocean depth (and, to a lesser extent, the sea-surface height $\eta$). The minimum and maximum ocean depth permitted in WINDS is 25m and 5250 m respectively. For water depth of 25 m and $\eta = 0$ m, the vertical resolution is 0.40 m at the surface, 0.67 m at the sea-floor, and the coarsest vertical resolution within the water column is 0.74 m. For water depth of 5250 m, the vertical resolution is 2.07 m at the surface, 280 m at the sea-floor, and the coarsest vertical resolution within the water column is 353 m. As a result, WINDS provides excellent vertical resolution within the upper water column, particularly in shelf seas where coral reefs are.

## 2.3 Bathymetry

We use GEBCO 2019 (GEBCO Compilation Group, 2019) as the basis for the bathymetry in WINDS. The nominal horizontal resolution of GEBCO 2019 is 15 arc-seconds (approximately 500 m) but, due to the lack of in-situ bathymetry measurements in the southwestern Indian Ocean, most bathymetry in this region is satellite-derived, with a practical resolution of around 6km (Tozer et al., 2019). Although these satellite-derived measurements are relatively well validated, there are problems in areas of extensive continental shelves and steep bathymetry (Tozer et al., 2019). These problems are quite dramatic in the southwestern

Indian Ocean. For instance, through comparison with Admiralty hydrographic navigation charts with in-situ soundings, we found local bathymetry errors in excess of 1 km around Aldabra Atoll, Seychelles, and a large number of erroneous 'islands' across Seychelles, which are in reality in significant water depth. Unfortunately, the only real solution to this lack of data is obtaining more in-situ bathymetric readings (e.g. see https://seabed2030.org/). However, to somewhat mitigate the most extreme errors in the southwestern Indian Ocean, we carried out two preprocessing steps of the GEBCO 2019 dataset. We firstly digitised all point-depth soundings from Admiralty Chart 718 (Islands North of Madagascar), including Aldabra, Assomption, Cosmoledo, Astove, and the Glorioso Islands, and then linearly gridded these data-points onto a regular 15 arc-second grid, carrying out necessary tidal adjustments, before linearly blending these grids with the rest of the GEBCO 2019 grid across a length-scale of 10-30 km. Secondly, to remove fake 'islands', we generated a land-sea mask at the GEBCO 2019 resolution from the highest resolution version of the GSHHG shoreline database (Wessel and Smith, 1996). We then set the depth of all false land cells (i.e. land according to GEBCO, ocean according to GSHHG) to 25 m. To avoid discontinuities in bathymetry, we applied a smooth $\tanh$ ramp between 25-50 m to all 'true' ocean cells shallower than 50 m (i.e. the shallower the bathymetry above 50 m, the more strongly the bathymetry would be nudged towards 50 m, with all bathymetry shallower 25 m shifted to deeper than 25 m). Although this minimum depth of 25 m is not realistic, it is a considerable improvement over a large number of fake islands, and a minimum depth of around 25 m is required for numerical stability at this resolution by CROCO anyway. As a final processing step, we carried out smoothing of bathymetry using `CROCO_TOOLS`, with a target $\nabla h / h = 0.25 \text{ m}^{-1}$, to improve model stability, and reduce pressure-gradient errors in regions of steep bathymetry. The bathymetry and associated grid parameters used in all WINDS simulations can be found in the `croco_grd.nc` file in the associated datasets, and is shown in Figure 1.

## 2.4 Experiments: WINDS-C and WINDS-M

WINDS is forced at the surface through a bulk formulation based on ERA-5 Hersbach et al. (2020), and at the lateral boundaries with the $1/12°$ GLORYS12V1 global ocean reanalysis (Lellouche et al., 2021) and tides. To investigate the importance of interannual variability in circulation in the southwestern Indian Ocean, we ran two experiments within WINDS. The first, WINDS-C, is based on a monthly climatology computed from ERA-5 and GLORYS12V1 from 1993-2019. The second, WINDS-M, is based on hourly forcing from ERA-5 and daily forcing from GLORYS12V1 from 1993-2019, plus an additional year (2020) based on the associated $1/12°$ global ocean analysis (the reanalysis product for 2020 was not available at this point). It is important that WINDS-M spans multiple decades, to fully incorporate the effects of multidecadal variability in surface circulation, and therefore dispersal (Thompson et al., 2018). WINDS-C was run after a 4-year spin-up, and WINDS-M was run from the end state of WINDS-C. WINDS-C and WINDS-M are otherwise identical.

## 2.5 Surface forcing

Surface forcing is parameterised using a bulk formulation based on the ERA-5 global atmosphere reanalysis (Hersbach et al., 2020) at hourly (WINDS-M) or monthly climatological (WINDS-C) temporal resolution, using the following fields, bilinearly interpolated to the WINDS grid:

- Surface air temperature (`t2m`)

- Sea-surface temperature (`sst`)

- Sea-level pressure (`msl`)

- 10m wind speed (`u10`, `v10`)

- Surface wind stress (`metss`, `mntss`)

- Specific humidity (`q`)

- Relative humidity (`r`)

- Precipitation rate (`mtpr`)

- Shortwave radiation flux (`msnswrf`)

- Longwave radiation flux (`msnlwrf`)

- Downwelling longwave radiation flux (`msdwlwrf`)

Unit conversions are required for most of these quantities to put them into the form used by CROCO. Since ERA-5 is computed on a different (coarser) grid to WINDS, there is a land-sea mask mismatch between ERA-5 and WINDS. To avoid terrestrial values erroneously being applied to ocean cells in WINDS, we masked out land values from ERA-5 using the ERA-5 land-sea mask, and carried out a nearest-neighbour interpolation over the small number of coastal WINDS cells that are counted as land cells in ERA-5.

## 2.6 Lateral forcing

### 2.6.1 Ocean currents

WINDS is forced at the lateral boundaries with the $1/12°$ GLORYS12V1 global ocean reanalysis, using daily-mean (WINDS-M) or monthly-climatological (WINDS-C) depth-varying ocean current velocities, sea-surface height, temperature, and salinity. GLORYS12V1 was run using tides and, as a result, we do expect there to be aliased tidal signals remaining in the daily-mean sea-surface height fields. However, we computed that the amplitude of the strongest aliased tidal signals (both SSH and currents) should be at least $20\times$ smaller than the true tidal signals, and frequency shifted to a period of 10-30 days. As a result, we do not expect that any remnant tidal signals in GLORYS12V1 will have any significant effect on tides in WINDS.

### 2.6.2 Tides

WINDS is forced at the lateral boundaries with 10 tidal constituents (barotropic tidal currents and surface height) from the TPXO9-atlas (Egbert and Erofeeva, 2002): $M_2$, $S_2$, $N_2$, $K_2$, $K_1$, $O_1$, $P_1$, $Q_1$, $Mf$, and $Mm$.

## 2.7   Rivers

We have simplistically included 12 major rivers in WINDS: the Zambeze, Rufiji, Tsiribihina, Mangoky, Ikopa, Betsiboka, Tana, Mahavavy Nord, Sambirano, Manambolo, Mananjary, and Ruvu rivers. We assume that water in the river-mouth area has a constant temperature of 25°C and a salinity of 15PSU, with monthly climatological discharge set according to Dai and Trenberth (2002). These riverine fluxes enter the ocean through the nearest ocean cell to the river mouth, set through inspection from satellite imagery (Google Earth). The location and annual-mean discharge of these rivers is shown in Figure 1.

## 3   Data Records

We have made three sets of output available from WINDS (Vogt-Vincent and Johnson, 2022a, b):

- minute output frequency

    - Zonal surface velocity (`u_surf`)

    - Meridional surface velocity (`v_surf`)

- 1 day output frequency

    - Sea-surface temperature (`temp_surf`)

    - Sea-surface salinity (`salt_surf`)

    - Free-surface height (`zeta`)

    - Depth-averaged zonal velocity (`u_bar`)

    - Depth-averaged meridional velocity (`v_bar`)

    - Kinematic wind stress (`wstr`)

    - Surface zonal momentum stress (`sustr`)

    - Surface meridional momentum stress (`svstr`)

    - Surface freshwater flux, E-P (`swflx`)

    - Surface net heat flux (`shflx`)

    - Net shortwave radiation at surface (`radsw`)

    - Net longwave radiation at surface (`shflx_rlw`)

    - Latent heat flux at surface (`shflx_lat`)

    - Sensible heat flux at surface (`shflx_sen`)

- 5 day output frequency

- Zonal velocity (`u`)

- Meridional velocity (`v`)

- Temperature (`temp`)

- Salinity (`salt`)

We did not output the vertical velocity. This can in principle be reconstructed at a 5 d frequency using the ocean depth, free-surface height, and zonal and meridional velocities.

## 4    Technical Validation

The following validation relates to WINDS surface properties only, as relevant for marine dispersal, since this was the primary use case WINDS-M and WINDS-C were run for. WINDS may, of course, be used for other purposes as well, but for these applications the model is provided *as is*. This validation focuses on WINDS-M, since WINDS-C is a control simulation which is not expected to fully reproduce observations as it is driven by low-frequency (monthly) climatological forcing.

### 4.1    Tides

We extracted the 5 largest tidal constituents ($M_2$, $S_2$, $N_2$, $K_1$, and $O_1$) at 50 sites across the WINDS domain (41 coastal, and 9 open-ocean) based on a 55-day 2-hourly time-series from WINDS-M 1994, and compared these amplitudes to the corresponding amplitudes in TPXO9-atlas (Egbert and Erofeeva, 2002), see Table S1. Note that this comparison is *not* independent since the TPXO9-atlas is used to set tidal boundary conditions at the WINDS domain boundaries. Additionally, TPXO9-atlas is not a purely observational product: it is a $1/30°$ inverse model constrained by observations. However, TPXO9-atlas is exten-
sively validated, and good agreement between tides in WINDS and TPXO9 does at least suggest that WINDS is propagating TPXO9-atlas tides reasonably.

Agreement between WINDS and TPXO9 is generally good, with tidal amplitude mismatch on the order of a few centimetres for almost all sites (well within the error associated with the TPXO9-atlas itself). A few regions associated with greater WINDS-TPXO9 disagreement include (1) the Sofala Bank (Mozambique) and (2) the mainland-facing sides of Mafia and
Zanzibar Islands (Tanzania). Both are shelf regions with extensive shallow water and, in the case of Tanzania, complex effects from nearby islands. The roughness length scale used in the bottom friction parameterisation in WINDS is constant, and the true ocean depth at these locations is occasionally shallower than the minimum depth used in WINDS, so it is possible that a combination of these two factors could explain the poorer tidal performance of WINDS in some shelf seas.

We have also carried out a comparison of WINDS tidal predictions with selected in-situ tidal gauges spanning the lon-
gitudinal and latitudinal range of WINDS, at Mombasa (Kenya), Aldabra (Outer Islands, Seychelles), Mahé (Inner Islands, Seychelles), Diego Garcia (Chagos Archipelago), and Mauritius and Rodrigues (Mauritius) (Table 1). This comparison demonstrates that WINDS can reproduce in-situ tidal predictions well, particularly at remote islands away from extensive continental shelves.

| Site/Constituent | Amplitude (cm, WINDS) | Amplitude (cm, observed) |
|---|---|---|
| **Mombasa (Kenya)** | | |
| $M_2$ | 102.8 | 105.5 |
| $S_2$ | 46.7 | 52.1 |
| $N_2$ | 17.8 | 20.1 |
| $K_1$ | 20.8 | 19.1 |
| $O_1$ | 10.5 | 11.3 |
| **Aldabra (Seychelles)** | | |
| $M_2$ | 94.0 | 93.3 |
| $S_2$ | 47.4 | 46.0 |
| $N_2$ | 16.5 | 17.4 |
| $K_1$ | 16.4 | 16.3 |
| $O_1$ | 8.9 | 10.0 |
| **Mahé (Seychelles)** | | |
| $M_2$ | 41.8 | 40.7 |
| $S_2$ | 19.6 | 18.1 |
| $N_2$ | 8.4 | 8.7 |
| $K_1$ | 18.6 | 18.7 |
| $O_1$ | 9.1 | 10.7 |
| **Diego Garcia (Chagos)** | | |
| $M_2$ | 47.6 | 49.3 |
| $S_2$ | 28.2 | 28.5 |
| $N_2$ | 8.7 | 8.9 |
| $K_1$ | 3.6 | 3.8 |
| $O_1$ | 3.3 | 3.9 |
| **Rodrigues (Mauritius)** | | |
| $M_2$ | 41.3 | 40.0 |
| $S_2$ | 22.8 | 25.5 |
| $N_2$ | 8.1 | - |
| $K_1$ | 5.9 | 5.0 |
| $O_1$ | 3.3 | - |
| **Mauritius (Mauritius)** | | |
| $M_2$ | 25.6 | 26.0 |
| $S_2$ | 14.2 | 15.8 |
| $N_2$ | 5.2 | - |
| $K_1$ | 6.0 | 6.1 |
| $O_1$ | 2.5 | - |

**Table 1.** Observational sources: Pugh (1979) (Mombasa, Aldabra, and Mahé); Lowry et al. (2009) (Rodrigues and Mauritius); Dunne (2021) (Diego Garcia).

## 4.2 Surface currents

Figures 2–4 compare monthly climatological surface currents averaged across 1993–2020 from WINDS (left); surface currents from 1993–2020 from Copernicus GlobCurrent, combining altimetric geostrophic currents with modelled Ekman currents (centre, Rio et al. (2014)); and near-surface currents estimated from the Global Drifter Program (GDP) using drifter trajectories from 1979–2015 (right, Laurindo et al. (2017)). Both products used for comparison are entirely independent of WINDS. These figures demonstrate that WINDS successively captures the location and velocity associated with major ocean currents in the southwestern Indian Ocean such as the Southern Equatorial Current, Southern Equatorial Countercurrent, North Madagascar Current, East Madagascar Current, and East African Coastal Current (e.g. Schott et al., 2009), as well as their seasonal variability. For instance, WINDS reproduces the observed strengthening of surface currents associated with the North Madagascar Current during the southeast monsoon (June-August), which can instantaneously reach approach $2 \, \mathrm{m \, s^{-1}}$, in accordance with in-situ observations (Swallow et al., 1988; Voldsund et al., 2017). The East African Coast Current (EACC) also correctly strengthens dramatically during the southeast monsoon, also reaching speeds of up to (and sometimes exceeding) $2 \, \mathrm{m \, s^{-1}}$, in agreement with observations (Swallow et al., 1991; Painter, 2020). The strongest surface currents in the EACC are simulated by WINDS to be close to the equator, and can instantaneously reach $3 \, \mathrm{m \, s^{-1}}$. We are not aware of observational evidence supporting such strong surface currents within the EACC. There is a discrepancy between the strength of surface currents simulated by WINDS and predicted by GlobCurrent for the South Equatorial Countercurrent close to the equator (e.g. see Figure 4, November). However, this is unsurprising as GlobCurrent uses geostrophic currents, which are not defined at the equator. Agreement between WINDS and GDP-derived surface velocities are much better in this region, with WINDS reproducing observations of zonal surface currents in excess of $1 \, \mathrm{m \, s^{-1}}$, particularly towards the east of the WINDS domain (Schott and McCreary, 2001; Shao-Jun et al., 2012).

To assess the ability of WINDS to reproduce surface current variability associated with eddies, Figure 5 compares the eddy kinetic energy (EKE) in WINDS and Copernicus GlobCurrent (high-frequency surface currents are not available from the Global Drifter Program), as well as 8 moorings from the RAMA array (McPhaden et al., 2009; Beal et al., 2019). The spatial pattern in EKE is similar between WINDS and Copernicus GlobCurrent, with both products returning high EKE associated with mesoscale eddy activity in the Mozambique Channel, around Mauritius and Réunion, in the wake of the Mascarene Plateau, and near the equator. EKE is generally higher in WINDS than Copernicus GlobCurrent, although this is likely in part due to sub-mesoscale turbulence simulated by WINDS, which will not be captured by Copernicus GlobCurrent. Compared to in-situ observations at RAMA array moorings, WINDS and Copernicus GlobCurrent tend to respectively overestimate and underestimate EKE. The RAMA time-series is considerably shorter than WINDS-M, and most moorings do not record equal coverage across the seasonal cycle. However, there does not appear to be a strong seasonal cycle in EKE in most regions (Figures S1-S3), so it is unlikely that this explains the systematically higher EKE in WINDS compared to RAMA. The currents measured at RAMA are also measured at a slightly greater depth (10/12m) than WINDS (0-2m). Nevertheless, this does suggest that eddies may be too energetic in WINDS. On the other hand, the variability of daily sea-surface height (Figure 6), and

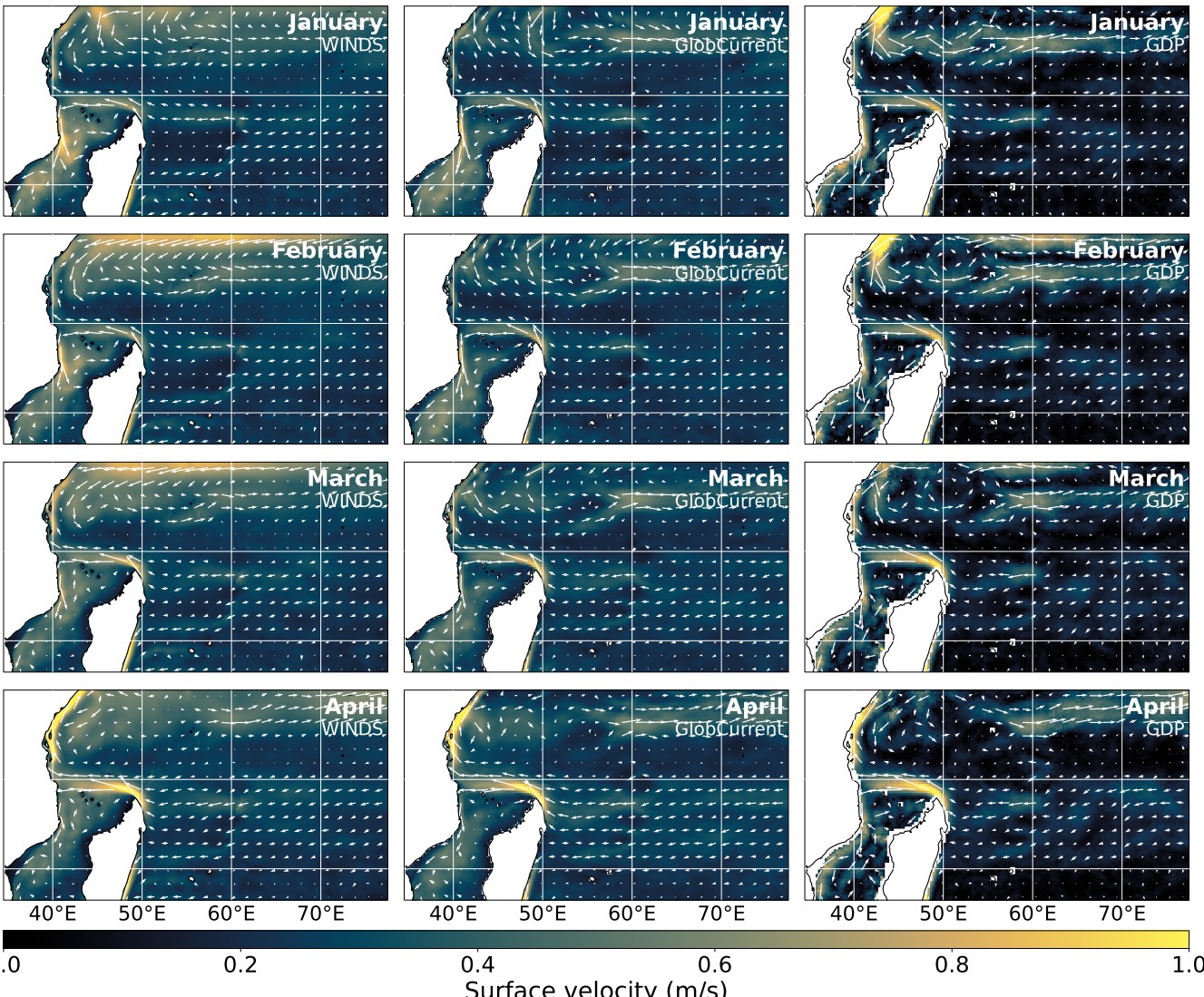

**Figure 2.** Monthly climatological surface currents (1993-2020) from WINDS (left), Copernicus GlobCurrent Surface (centre), and Global Drifter Program derived near-surface currents (right) for January to April.

therefore geostrophic surface currents, agrees very well with observations. This suggests that, at least away from the equator, mesoscale eddy activity is reasonably reproduced in WINDS.

The monthly mean surface current speed in WINDS associated with major surface currents in the southwestern Indian Ocean is shown is Figure 7 (see Figure S8 for geographical reference), compared to a 1/12° global ocean reanalysis (CMEMS GLO-RYS12V1) and Copernicus GlobCurrent. Particularly strong agreement between GLORYS12V1 and GlobCurrent is expected

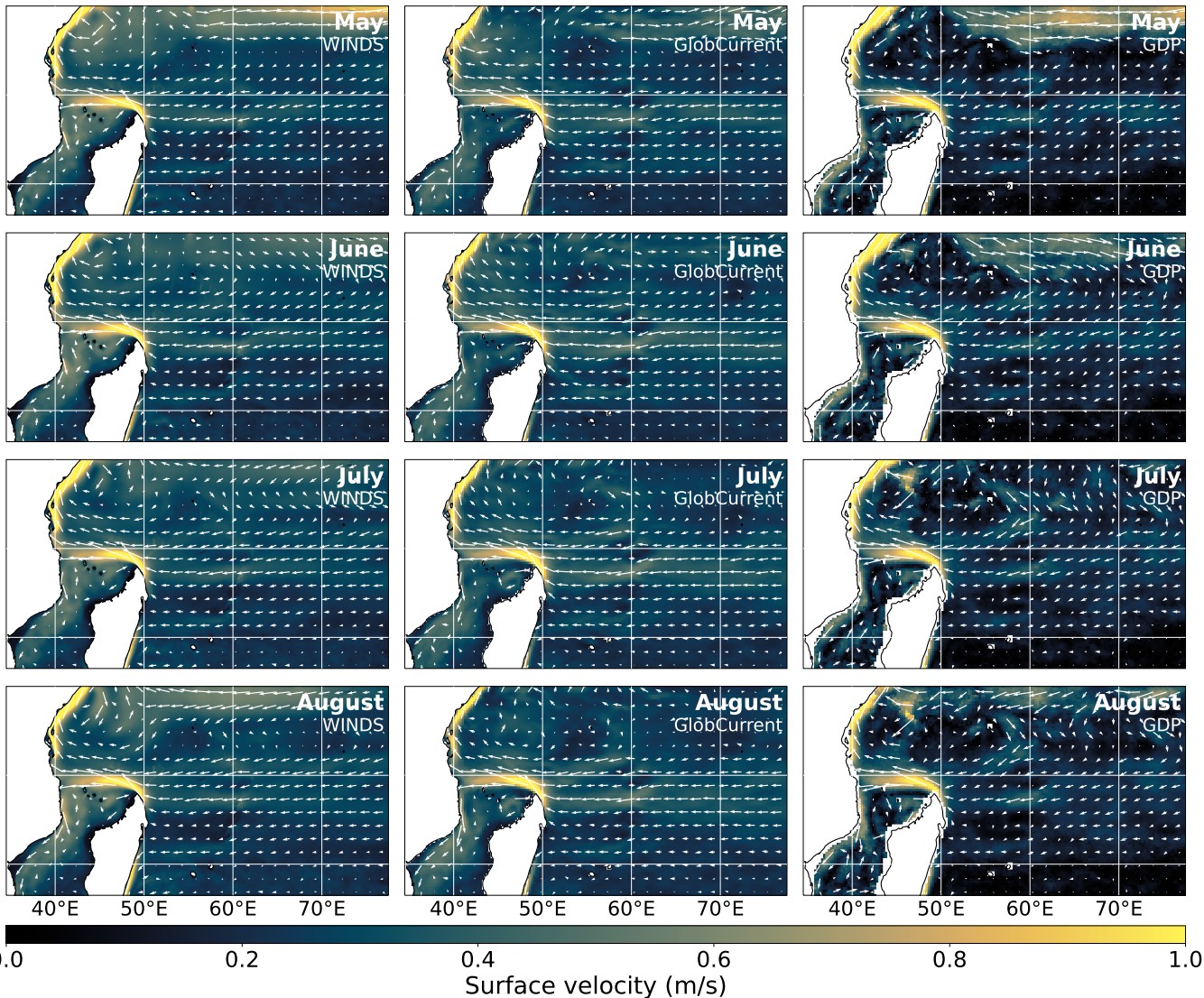

**Figure 3.** Monthly climatological surface currents (1993-2020) from WINDS (left), Copernicus GlobCurrent Surface (centre), and Global Drifter Program derived near-surface currents (right) for May to August.

as the former is an assimilative model, but agreement is generally very good between all three products. One notable exception is the western South Equatorial Countercurrent, where current variability often appears to be greater in WINDS than GLO-RYS12V1 and GlobCurrent (which is aso reflected in EKE derived from RAMA moorings in Figure 5). The seasonal cycle is also amplified in the NW Mozambique Channel in WINDS compared to GLORYS12V1 and GlobCurrent. Very high surface current speeds have been observed in this region from in-situ observations (Ridderinkhof et al., 2010), but it is not clear whether the stronger seasonality simulated by WINDS is real or an artefact. Although the seasonal monsoonal cycle dominates many

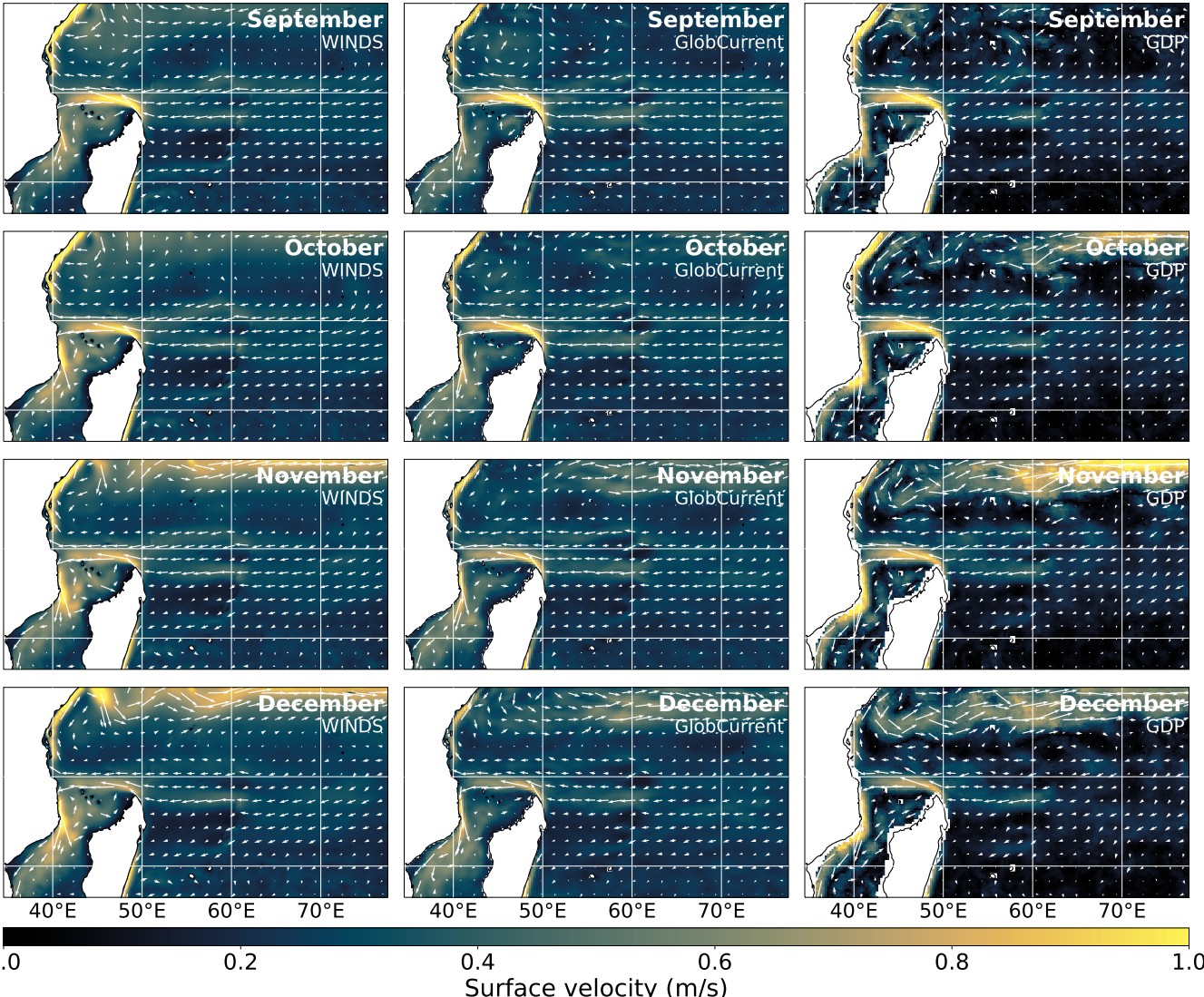

**Figure 4.** Monthly climatological surface currents (1993-2020) from WINDS (left), Copernicus GlobCurrent Surface (centre), and Global Drifter Program derived near-surface currents (right) for September to December.

of the time-series in Figure 7, it is also clear that there is considerable interannual variability. This is generally reproduced very well by WINDS, but the magnitude of interannual variability is occasionally larger in WINDS than GLORYS12V1 or GlobCurrent (see also Figure S9).

     As WINDS was designed for the primary purpose of simulating marine dispersal (for instance, for coral larvae), it is important to test whether WINDS can reproduce observed pathways of surface drift in the ocean. Although Global Drifter Program

(GDP) deployments are low in the southwestern Indian Ocean (Lumpkin and Centurioni, 2019), sufficient drifters have passed

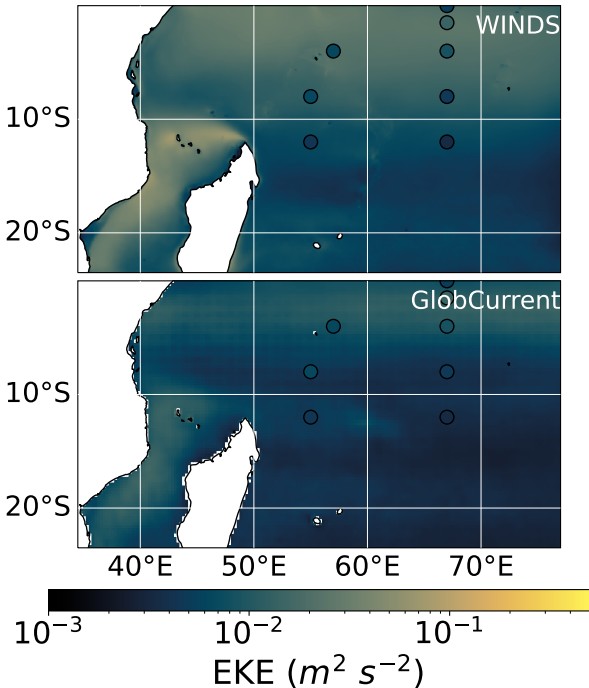

**Figure 5.** Eddy kinetic energy (EKE) from WINDS (top) and Copernicus GlobCurrent (bottom). EKE was computed by passing daily-mean surface velocity through a high-pass filter with a cutoff period of 30 days, thereby removing high-frequency variability associated with tides, and low-frequency variability associated with time-mean currents and the seasonal cycle. Circles represent the EKE at 10/12 m depth from the RAMA array. EKE is also plotted as a monthly climatology in Figures S1-S3, and MKE (annual-mean and monthly-climatological) in Figures S4-S7.

through the region to evaluate first-order dispersal pathways simulated by WINDS. We released a large number of virtual Lagrangian particles at coral reef sites around 6 islands and banks within the WINDS domain on the 1st, 11th, and 21st of each month from 1993-2019, and advected them for 120 days following WINDS-M surface currents using OceanParcels (Lange and Sebille, 2017), with a Runge-Kutta 4th order scheme and a time-step of 10 minutes. Figure 8 shows the proportion of
these virtual particles that passed through each $0.5° \times 0.5°$ grid cell at least once within 120 days, overlaid with GDP drifter trajectories for (up to) 120 days after their nearest approach to release sites. Although the Global Drifter Program sample size is small in some cases, agreement between observed drifter trajectories and WINDS is generally excellent, with observed trajectories usually confined to the 'high-probability' regions predicted by WINDS. Notable exceptions include some GDP drifters that travelled further eastward within the South Equatorial Countercurrent from Mayotte and Zanzibar than predicted
by WINDS, and less zonal confinement within the South Equatorial Current for GDP drifters travelling westwards from the Chagos Archipelago as compared to WINDS. These trajectories are physically possible within WINDS (Figure S10), but improbable. Surface currents associated with the South Equatorial Countercurrent in WINDS are generally at least as strong

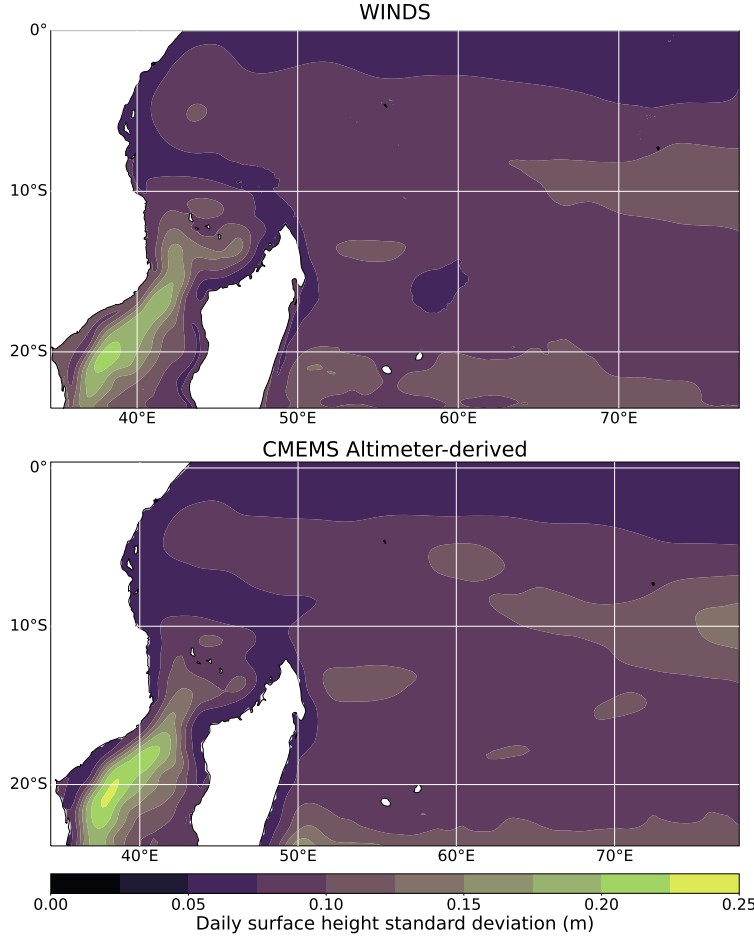

**Figure 6.** Variability of sea-surface height from 1993-2020 from WINDS (top) and the CMEMS Global Ocean Reprocessed Gridded L4 sea-surface Height product (bottom), computed as a standard deviation.

as those diagnosed from GDP drifters (Figure 2-4), so the possible underprediction of eastward surface transport is instead likely due to the WINDS domain ending at the equator. In our Lagrangian analysis, any virtual particles crossing the equator

are removed, so many virtual particles may be unable to enter the South Equatorial Countercurrent. For instance, most GDP drifters in Figure 8 that travelled a significant distance within the South Equatorial Countercurrent passed the equator at least once. It is nevertheless possible that WINDS underestimates surface connectivity between the East Africa Coastal Current and the South Equatorial Countercurrent, as one GDP drifter followed a relatively direct pathway from Zanzibar to the Seychelles Plateau, which was improbable in WINDS. It is important to note that GDP drifters are drogued and have some wind exposure

due to the buoy, and may therefore experience forces from winds, waves and subsurface currents which will not be captured by WINDS.

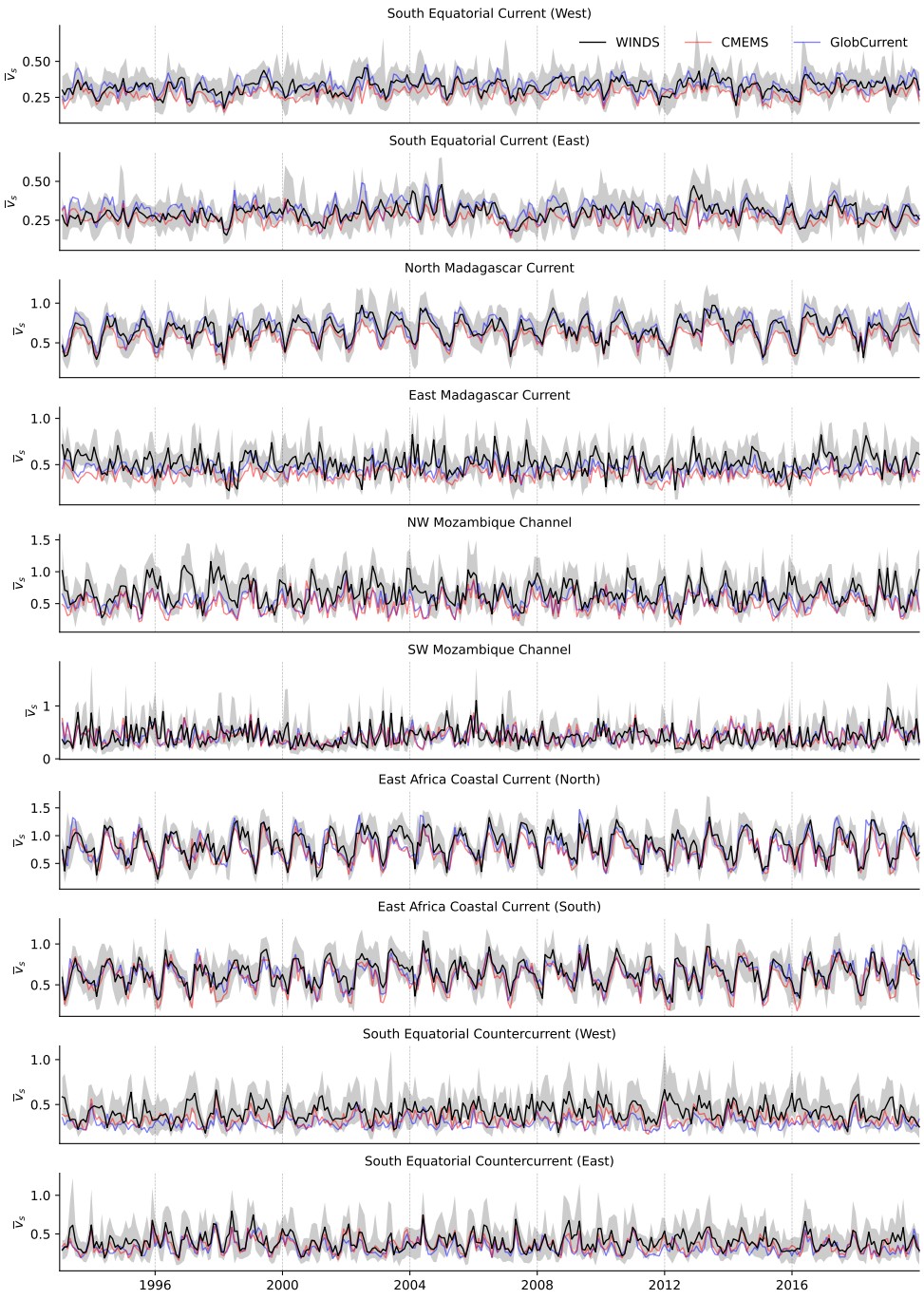

**Figure 7.** Monthly mean surface currents averaged across ten key regions (see Figure S8 for geographical reference) for WINDS (black, with grey shading representing the monthly range), CMEMS GLORYS12V1 (red), and GlobCurrent (blue).

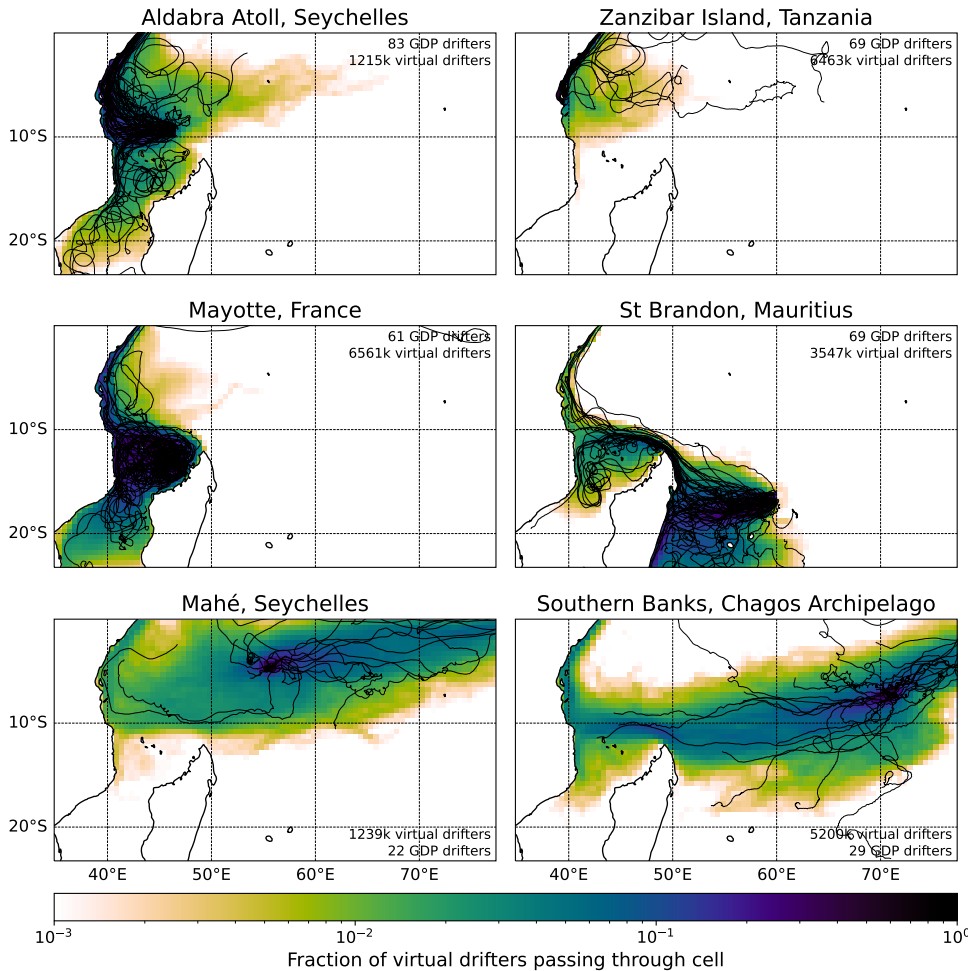

**Figure 8.** *Colour:* Fraction of virtual drifters advected with half-hourly WINDS-M surface currents that pass through each $0.5° \times 0.5°$ grid cell at least once within 120 days. Cells with less than 0.1% of virtual drifters passing through are shaded in white. *Lines:* Observed Global Drifter Program drifter trajectories, for 120 days following the nearest pass to the virtual drifter release site (or until beaching/death, if this occured within 120 days).

Long-distance dispersal patterns predicted by WINDS are similar to those predicted by the CMEMS GLORYS12V1 1/12° global ocean reanalysis (Figure S11), and, given the relatively small sample size of GDP drifters in the region, it is not clear which product performs better over these very large distances. Indeed, this is expected, as the 1/12° horizontal resolution of GLORYS12V1 is likely sufficient to resolve the main processes relevant to the large-scale surface circulation in the Indian Ocean. The real potential advantage of WINDS is at finer scales, which is particularly important for local applications, or for modelling the dispersal of substances with a lifespan on the order of days to weeks (rather than months or longer), such as coral larvae Connolly and Baird (2010). For instance, Figure 9 shows simulated trajectories of 25 virtual surface-confined

particles, perhaps representing coral larvae, released from analogous reef sites in southern Zanzibar, Tanzania, in WINDS-M and GLORYS12V1, at the same time on July $1^{st}$ 2019. Perfect agreement between the two is *not* expected as (i) WINDS is not assimilative and (ii) there was likely limited observational data available for assimilation in this region in the first place. However, it is clear from this figure that the representation of the coast and islands is significantly improved in WINDS relative to GLORYS12V1 (due to the more than four-fold increase in horizontal resolution). For instance, some virtual particles in WINDS entered Kiwani Bay in southwestern Zanzibar, rather than entering the East Africa Coastal Current. This is physically impossible in GLORYS12V1, as the resolution is too coarse to resolve all but the largest coastal features. WINDS also simulates greater particle-particle dispersion due to resolved motion that would be sub-grid scale for GLORYS12V1 (Poje et al., 2010). The approximately 2 km resolution of WINDS is still too coarse to resolve the finest scales of motion that are important for reef-scale disersal (Dauhajre et al., 2019; Grimaldi et al., 2022), but at intermediate scales of tens to hundreds of kilometres, we expect that WINDS will provide a significantly improved capacity for dispersal modelling and associated applications as compared to existing openly available regional and global ocean simulations.

## 4.3 Sea-surface temperature (SST) and salinity (SSS)

We have validated WINDS SST and SSS predictions by comparing monthly climatological SST and SSS from WINDS-M, to monthly climatological SST from OSTIA (Good et al., 2020) and SSS from ARMOR3D (Guinehut et al., 2012), all computed across 1993-2020 (both are independent of WINDS, although it is important to note that observations for sea-surface salinity are sparse in the southwestern Indian Ocean). In general, WINDS performs well for both SST and SSS; the mean absolute error (MAE) for SST and SSS respectively ranges between 0.14–0.24 °C, and 0.06–0.1 PSU across the seasonal cycle (Figures 10 and S12). There is a widespread and year-round cold and fresh bias across most of the southwestern Indian Ocean in WINDS, although the magnitude of this bias is small. There is also a warm bias within the Mozambique Channel during the northwest monsoon (November to February), and a salty bias year-round. We do not know for certain why these biases exist in WINDS, although it may be related to the GLS vertical mixing parameterisation, resulting in an over/underestimate of the mixed-layer depth. WINDS also appears to slightly overestimate the strength of the seasonal SST cycle in shallow water along the coasts of East Africa and Madagascar. Finally, there is a spatially limited but relatively intense fresh bias in WINDS off the coast of Mozambique, associated with the Zambeze River. The implementation of rivers in WINDS is simplistic, so it is possible that the seasonal discharge climatology or physical water properties associated with the river mouth (15 PSU and 25 °C) were inappropriate, that the advection of the freshwater plume associated with the river is incorrectly simulated in WINDS, or that ARMOR3D does not fully capture the fine-scale freshwater plume associated with the Zambeze River. Figure S13 shows time-series of the difference in SST and SSS between WINDS, and OSTIA and ARMOR3D, across the simulation timespan. Errors in both SST and SSS follow a seasonal cycle (as indicated by Figures 10 and S12) but annual mean SST errors are relatively consistent from 1993-2020. There is a reduction in errors associated with salinity after 2004, however, perhaps due to improvements in the availability of observations for data-assimilation in ERA-5 (setting ocean-atmosphere fluxes in WINDS).

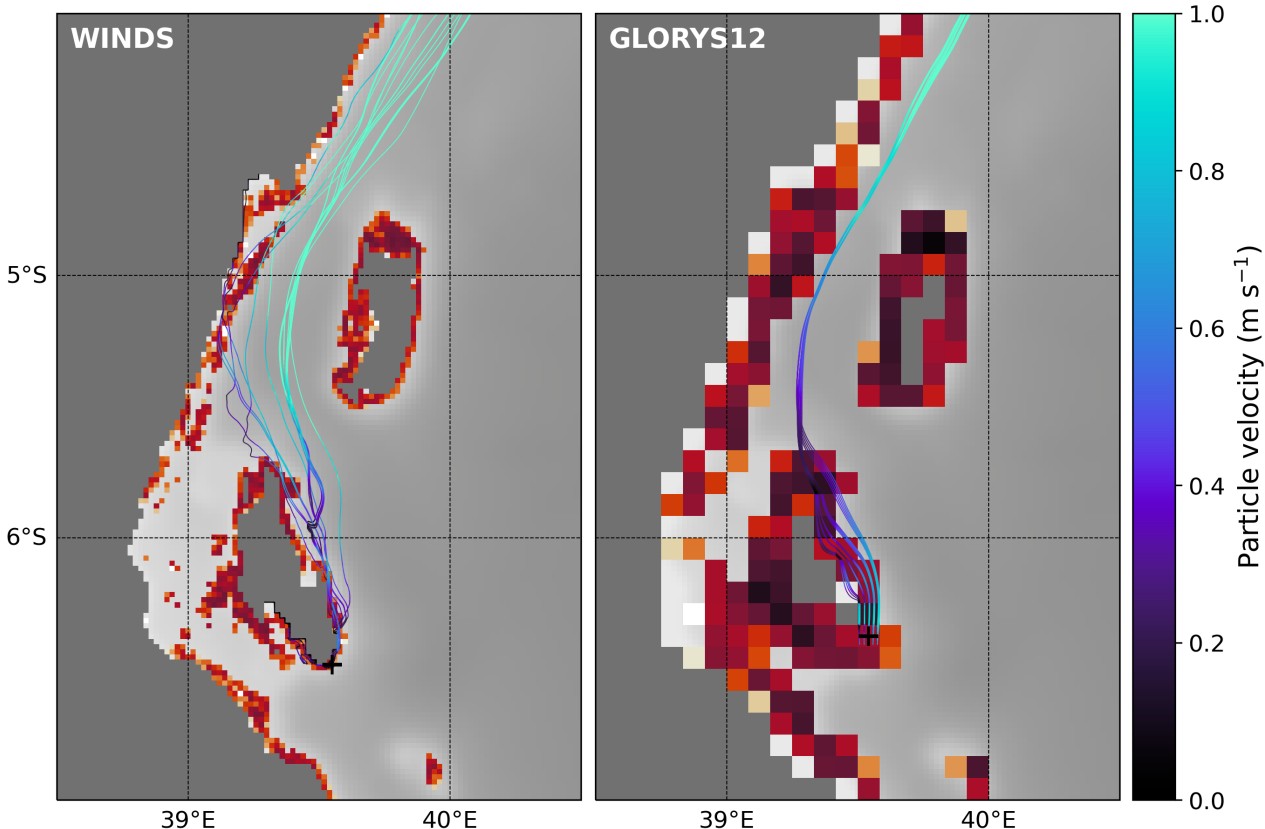

**Figure 9.** 25 virtual drifter trajectories, coloured by instantaneous speed, released from analogous coral reef cells at the southern tip of Zanzibar (Tanzania) on July 1$^{st}$ 2019 in WINDS-M (left) and GLORYS12V1 (right). Land cells for both models are shaded in dark grey. Red cells are coral reefs (Li et al., 2020), aggregated to the respective model grid, and shaded by total area per cell (shown for illustrative purposes only).

## 5   Conclusions

WINDS, and specifically the realistic WINDS-M experiment, reproduces surface circulation well in the southwestern Indian Ocean. Although surface current variability may be overestimated by WINDS in certain regions, such as within 5° of the equator, WINDS-M successfully reproduces the main features of surface circulation across the region, observed surface drifter pathways, and surface properties such as temperature and salinity. Although observations of sub-mesoscale circulation in particular in the southwestern Indian Ocean are lacking, our validation of WINDS-M suggests that this product is suitable for model-based studies investigating the dispersal of buoyant particles on scales of $\mathcal{O}(10^1 - 10^3)$ km. To our knowledge, the spatial resolution of WINDS-M is a four-fold improvement on the highest resolution publicly available time-varying dataset for surface currents in the southwestern Indian Ocean (1/12° global ocean (re)analyses, such as GLORYS12V1 (Lellouche

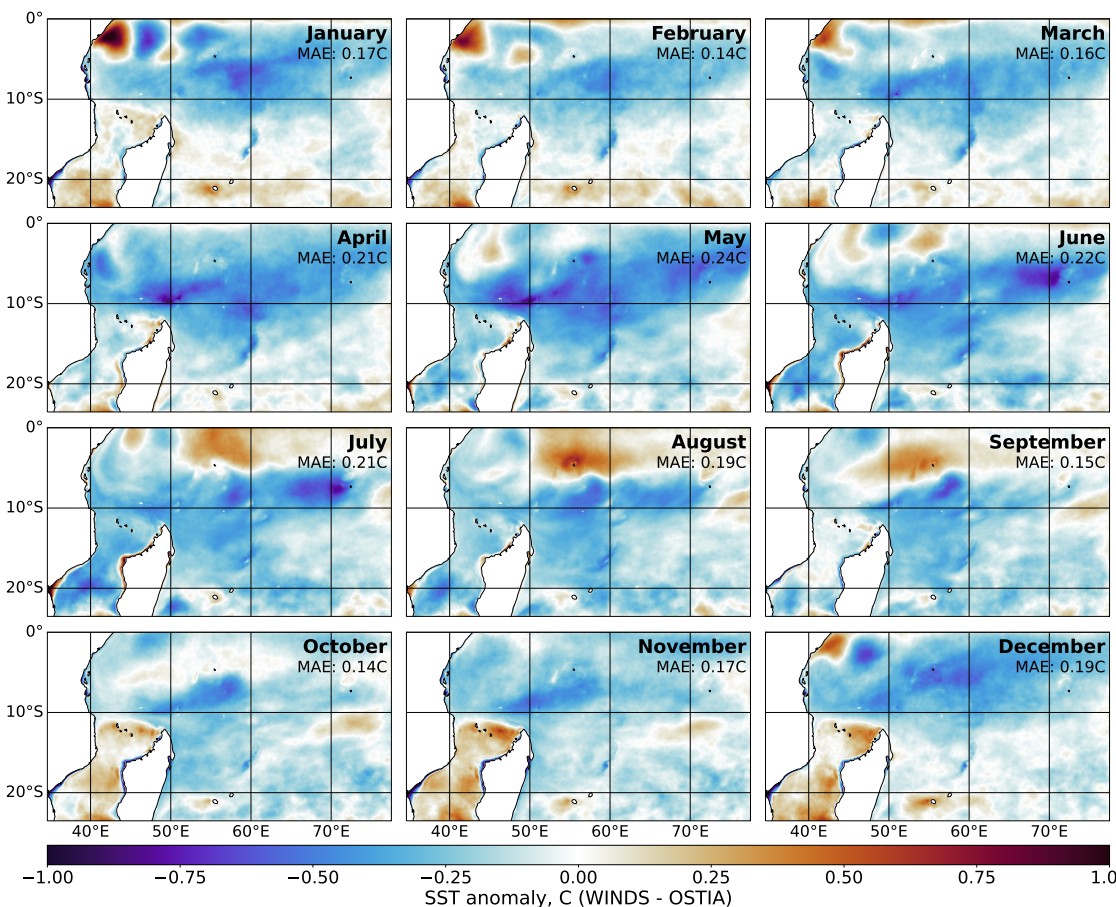

**Figure 10.** Difference between monthly climatological SST simulated by WINDS, and satellite and in-situ derived SST estimates from OSTIA. Blues indicate that WINDS simulates cooler temperatures, reds indicate that WINDS is warmer.

et al., 2021)), and the temporal resolution (30 minutes) is also sufficient to capture a wide range of current variability. We hope that the output of WINDS will be useful for those investigating marine dispersal (and, more broadly, marine science) in the southwestern Indian Ocean.

*Code and data availability.* The full dataset (WINDS-C and WINDS-M), as summarised in section 3, is permanently archived at the British
Oceanographic Data Centre (Vogt-Vincent and Johnson, 2022a, b):

– **WINDS-C**: http://dx.doi.org/10.5285/b2b9bfe408f14ea7a79d9ff7aee0d0b8

– **WINDS-M**: http://dx.doi.org/10.5285/BF6F0CFBD09E47498572F21081376702

We have also provided the CROCO configuration files that were used to run WINDS, as well as the model grid, and forcing files used by WINDS-C (the forcing files used by WINDS-M were too large to store permanently, but are described in sections 2.5 and 2.6). The configu-

ration files and code required to reproduce figures in this manuscript are archived here (Vogt-Vincent, 2023).

CROCO V1.1 is available to download here, with the documentation archived here.

*Video supplement.* Supplementary video 1: visualisation of 1 year of surface temperatures from WINDS-C Year 8, at daily resolution, generated for outreach purposes. Surface temperature is rendered as a heightmap for this visualisation to highlight flow, and the colourmap
range is 22-30°C.

*Author contributions.* NV: *Conceptualisation, methodology, software, validation, writing (original draft), visualisation, funding acquisition.* HJ: *Conceptualisation, methodology, resources, supervision, funding acquisition.*

*Competing interests.* The authors declare that they have no known competing interests that could have appeared to influence the work reported in this paper.

*Acknowledgements.* This work was funded by NERC grant NE/S007474/1, and used the ARCHER2 UK National Supercomputing Service (https://www.archer2.ac.uk) and JASMIN, the UK collaborative data analysis facility. CROCO and CROCO_TOOLS are provided by http://www.croco-ocean.org. The data analyses in this study made use of CDO (Schulzweida, 2022) and a range of python modules, including xarray (Hoyer and Hamman, 2017), scipy (Virtanen et al., 2020), cmasher (van der Velden, 2020), dask (Dask Development Team, 2016), matplotlib (Hunter, 2007), and OceanParcels (Lange and Sebille, 2017; Delandmeter and van Sebille, 2019). We are grateful for the time,
comments and suggestions from the two anonymous reviewers, which has significantly improved the clarity and utility of this paper.

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
