# Peer review of "Multidecadal and climatological surface current simulations for the southwestern Indian Ocean at 1/50° resolution"

_Geoscientific Model Development, 2022_

## Author Response (AR1)

**Response to Reviewer Comment 1**

We thank the reviewer for their time they have invested in reviewing our study, and we are grateful for their positive assessment of the manuscript. The reviewer recommended publication in GMD if the manuscript fits the journal scope, and did not make suggestions for improvement. We hope the reviewer will agree that the changes we have made to the manuscript in response to the other reviewer's comments have improved the manuscript.

**Response to Reviewer Comment 2**

We thank the reviewer for the time they invested in reviewing our manuscript, their positive evaluation of our study, and their thoughtful comments and suggestions. Please see below for a response to their three comments:

**The title says multi-decadal and climatological simulations, however the evaluation is only done at a climatological scale. Is there a way to evaluate any multidecadal variability of the surface characteristics being analyzed here?**

We agree with the reviewer that this should be improved. We have created a new figure (Figure 7 in the revised manuscript) in which we show the mean monthly surface current speed averaged across ten regions (mostly 2°×2°, with the exception of the South Equatorial Current and Countercurrent, which are both rather broad currents) within the WINDS domain, see Figure S8 in the revised Supplementary Materials for a geographic reference. These ten regions collectively represent all of the major ocean currents within the region (the North and Eastern Madagascar Currents, the South Equatorial Current and Countercurrent, the East Africa Coastal Current, and boundary currents in the Mozambique Channel). We have compared these average surface speeds to (i) Copernicus GlobCurrent (i.e. geostrophic and Ekman currents computed from observed sea-surface altimetry and winds), and (ii) the CMEMS GLORYS12V1 1/12° global ocean reanalysis. We have also added a figure to the Supplementary Materials (Figure S9) with the same analysis passed through a low-pass filter with a cut-off frequency of 1/480 days, to isolate interannual current variability. As described in a new paragraph in the revised manuscript, there is generally very good agreement between WINDS-M and both GLORYS12V1 and GlobCurrent. In some regions, WINDS-M predicts greater interannual variability relative to GLORYS12V1 and GlobCurrent. Due to a lack of openly available in-situ observations, we are currently unable to verify whether this is a genuine overestimate or not.

However, in general, agreement is good, and Figure 7 in the revised manuscript will allow users to assess whether interannual variability of currents in their particular region of interest is in strong or weak agreement with estimates from an established global ocean reanalysis and satellite-derived models. For instance, we would be comfortable in using WINDS-M to investigate interannual variability associated with the East Africa Coastal Current, whereas more caution may be needed for variability within the Mozambique Channel.

**The choice of the southern boundary of the domain seems to be quite random. If the main focus, as mentioned in the introduction, is not only the Seychelles but also the southwestern Indian Ocean as whole, then the southern boundary is placed further north. On the other hand, if the focus is only on the Seychelles, then the choice of such a large domain is a bit wasteful.If focusing on the southwestern Indian Ocean, there are few things to be considered. Just to name a few there is the Southeast Madagascar current along with southeast Madagascar bloom (e.g. Dilmahamod et al., 2020), which are important features in the area. Similarly, the Sofala bank over the Mozambique Channel where four rivers along the eastern African coast drain into the bank, plays important roles**

**on the dynamics of marine organisms in the area (e.g. Malauene et al., 2018). These features are important in the area but being cut by the southern boundary.**

We realise that we did not fully explain why we chose 23.5° to be the southern boundary. Although Seychelles was a project focus due to funding obligations, our overall primary aim was to assess coral reef connectivity across the southwestern Indian Ocean. We were also particularly interested in assessing connectivity between the Chagos Archipelago and the rest of the southwestern Indian Ocean. As a result, a priority was that WINDS should specifically cover as many coral reefs as possible. The Chagos Archipelago is a rather long distance away from all other reefs in the southwestern Indian Ocean so, to keep the Chagos Archipelago within the WINDS domain, we decided that compromises were needed along the southern border. The WINDS domain excludes a small number of coral reefs in southernmost Madagascar and South Africa, but it is otherwise very comprehensive. If we had known that potential users were particularly interested in a further southward extension of the WINDS domain prior to running the simulations, it may have been possible to consider this. However, this was unfortunately not raised in our stakeholder consultations (which were, admittedly, rather Seychelles-centric) and the WINDS dataset is already rather large (>10TB). The large size of this dataset means that carrying out Lagrangian dispersal experiments is already highly memory and bandwidth intensive, and this would be made even more difficult with a further domain expansion.

We hope the reviewer will understand this justification. To clarify our priorities in designing the WINDS configuration, we now explicitly mention our aim of including the Chagos Archipelago in the introduction, and de-emphasise Seychelles (since, although this was a priority for us, it will not be of relevance to most readers).

**The evaluation/ comparison with observational/reanalysis data is clear, however it would be helpful also if the authors could refer to how WINDS perform compared to other models ( ROMS – ex CROCO for instance) in the area. There are modeling studies, especially on the eddies over the Mozambique Channel, that have been conducted in the area before (e.g. Halo 2012 on the Mozambique Channel eddies). This could be an opportunity to showcase the importance of having high res model in the area?**

We completely agree with the reviewer that a comprehensive comparison with other models would be valuable. The importance of model resolution on dispersal characteristics has been investigated at coarser (e.g. Blanke *et al.* 2012; Qin *et al.* 2014) and finer (e.g. Dauhajre *et al.* 2019) horizontal resolutions, but we are not aware of a systematic analysis in the range of 1-10km. It was (originally, at least) part of the project plan to carry out such an analysis by using an identical larval dispersal experiment design in WINDS-M, GLORYS12V1, and GlobCurrent. However, this unfortunately goes beyond the scope of the present study.

We nevertheless agree with the reviewer that the (potential) advantages of WINDS relative to other models was not made clear in the manuscript. Most high-resolution models run in the region were never made publicly available, so the main alternatives to WINDS are GLORYS12V1 and HYCOM. Since the raison d'être for WINDS is dispersal modelling applications, we extended the model evaluation by carrying out dispersal experiments from six example coral reef sites within the WINDS domain, with virtual particles generated at each site three times per month from 1993-2019, and then advected for 120 days following WINDS-M surface currents. We then extracted Global Drifter Program (GDP) drifter trajectories for 120 days following the nearest pass to each reef site, and compared the two (Figure 8 in the revised manuscript). This in itself is a new useful evaluation for WINDS.

We then carried out the same experiment using GLORYS12V1 and, although there are some minor differences (relating to disagreement on low-probability trajectories, see Figure S11 in the revised Supplementary Materials), in general, both GLORYS12V1 and WINDS agree well with GDP drifter trajectories, and it is not possible to assess which product performs better due to (i) the relatively small GDP drifter sample size in the region, and (ii) physical reasons why GDP drifters may not perfectly follow surface currents. The general good agreement between GLORYS12V1 and WINDS for long-distance dispersal should be unsurprising, as the 1/12° resolution of GLORYS12V1 is probably perfectly adequate to reproduce the large-scale wind-driven surface circulation of the Indian Ocean.

The real advantages of WINDS, are for more local-scale (tens to hundreds of kilometres) processes. This will be particularly important for (i) marine practitioners working on local-scale problems (e.g. marine debris accumulation and attribution at a locality or island level) and (ii) assessing the dispersal of substances with a relatively low lifespan in the ocean, such as many types of larvae. It is for these specific applications that a comparison with coarser-resolution models in the region would be most valuable.

As indicated above, we think that such an in-depth analysis goes beyond the scope of the present study. In particular, although a model-to-model sensitivity study would be valuable, true validation would only be possible with in-situ observations that are sensitive to these fine-scale oceanographic processes (e.g. observations of differences in marine debris accumulation rate across around 100km of coastline would be a great way of evaluating this). As a compromise, to more clearly communicate the potential benefits of WINDS, we have added a new Figure 9 to the revised manuscript. In this figure, we plot 25 example trajectories of surface-confined Lagrangian particles (e.g. perhaps virtual coral larvae) from a reef in southernmost Zanzibar Island in WINDS and GLORYS12V1. The exact difference between the trajectories is irrelevant (since WINDS is not assimilative and there will be limited observations from the region feeding into GLORYS12 anyway) but this figure illustrates the following points:

- Visually emphasises the improvement in resolution of WINDS relative to the best existing openly available ocean models in the region.
- The significantly improved horizontal resolution of WINDS relative to GLORYS12V1 means that WINDS can simulate dispersal pathways that are fundamentally impossible in GLORYS12 (e.g. a number of trajectories ending up in a bay which isn't resolved by GLORYS12).
- WINDS may facilitate applications that are relevant to relatively fine scales (e.g. in this case, individual sections of coastline within Zanzibar).
- Surface transport in WINDS is more dispersive than in a coarser ocean model.

**References**

Blanke et al. (2012), 'Sensitivity of Advective Transfer Times across the North Atlantic Ocean to the Temporal and Spatial Resolution of Model Velocity Data: Implication for European Eel Larval Transport'.

Dauhajre, McWilliams, and Renault (2019), 'Nearshore Lagrangian Connectivity: Submesoscale Influence and Resolution Sensitivity'.

Qin, van Sebille, and Sen Gupta (2014), 'Quantification of Errors Induced by Temporal Resolution on Lagrangian Particles in an Eddy-Resolving Model'.

---

## Referee Report (RR1)

**Multidecadal and climatological surface current simulations for the southwestern Indian Ocean at 1/50º resolution**

The authors went through a thorough revision of the manuscript. They have added some clarification on the points made during the first round of review. The manuscript can be accepted as is.